# Advances in Methodology for Fatigue Assessment of Composite Steel–Concrete Highway Bridges Based on the Vehicle–Bridge Dynamic Interaction and Pavement Deterioration Model

**Ana Célia Soares da Silva [1], Guilherme Santana Alencar [2] and José Guilherme Santos da Silva [1,3,*]**

1    Civil Engineering Post-Graduate Program (PGECIV), State University of Rio de Janeiro (UERJ), São Francisco Xavier St. 524, Rio de Janeiro 20550-900, RJ, Brazil; anaceliasoares.eng@gmail.com

2    Civil and Environmental Engineering Department, University of Brasilia (UnB), Brasília 70910-900, DF, Brazil; guilherme.alencar@unb.br

3    Structural Engineering Department (ESTR), State University of Rio de Janeiro (UERJ), São Francisco Xavier St. 524, Rio de Janeiro 20550-900, RJ, Brazil

\*    Correspondence: jgss@uerj.br

**Abstract:** Fatigue cracking is one of the most prominent causes of mechanical failure limiting the service life of existing steel and composite steel–concrete bridges and is among the central concerns of structural and bridge engineers. In this context, the current work presents some recent advancements in an existing methodology for fatigue analysis developed by the authors throughout the years. The methodology is specifically devoted to the fatigue assessment of composite steel–concrete bridges employing the local hot-spot S-N approach and a coupled vehicle–pavement–bridge system considering progressive pavement deterioration with stochastically generated roughness profiles. Two different methodologies were used to solve the dynamic equilibrium equations: the modal superposition method to solve the bridge dynamic equations and a direct integration method to solve the vehicle dynamic equations. From a computational point of view, the present approach is more efficient and detailed than previous versions, as it allows a significant reduction in the analysis time and the use of complex bridge and vehicle finite element models. In this regard, a case study of a highway composite steel–concrete bridge spanning 40 m was selected in order to demonstrate the usefulness of the presented improved methodology by carrying out a fatigue analysis. The results of this investigation (displacements and stresses) are presented, aiming to verify the factors that directly influence the structural response and, consequently, the service life of steel–concrete composite highway bridges.

**Keywords:** dynamic structural analysis; finite element modelling; fatigue behaviour; hot-spot stress method

## 1. Introduction

Highway bridges are usually subjected to random dynamic actions of variable magnitude due to vehicles crossing on the bridge deck pavement throughout their service life. These dynamic actions can induce a significant increase in stress amplitudes, and consequently, serious problems related to the fatigue phenomenon can occur, such as the nucleation of fractures or even their propagation in bridge structural details. In this context, being able to carry out accurate estimates of the fatigue service life of a highway bridge structure is a highly valuable asset for bridge engineers dealing with bridge design and the assessment of existing infrastructure.

In this scenario, evaluating the dynamic effects of vehicles on steel and composite highway bridges becomes extremely necessary. However, bridge design has been traditionally carried out through static or quasi-static analyses, considering equivalent loads based on design standards with a dynamic amplification factor, which conservatively simulates

the dynamic effects of vehicle traffic [1,2]. Therefore, in recent years, many studies have been conducted with the aim of improving the provision of design standards.

Ludescher and Brühwiler [3] investigated the dynamic amplification of traffic in a simply supported composite highway bridge using a simplified bridge and vehicle model, both with one degree of freedom. Deng and Cai [4] performed calculations with the vehicle–structure interaction in order to analyse the influence of the span length, surface roughness and vehicle speed on the structural response of the bridge. The authors concluded that, for pavement with poor surface conditions, AASHTO [5] may underestimate the dynamic amplification effects.

Some authors, including Oliva et al. [6–8], performed an analysis of the roughness influence in terms of dynamic amplification values in a simply supported bridge and in a three-span continuous bridge. In this regard, the results demonstrated that the dynamic response was overestimated when the differences between the left and right roughness profiles were neglected in the case of 3D models with two profile lines corresponding to the vehicle circulation lines.

Zhao and Uddin [9] analysed the dynamic response obtained in experimental tests and compared these results with the equivalent static value, which was evaluated considering a numerical model. The authors concluded that, although the dynamic amplifications specified by the standards are compatible with the specific analysed case, further studies with different types of bridges and vehicles should be carried out to obtain a general conclusion.

In this context, Han et al. [10] investigated dynamic amplifications in typical highway bridges due to extra-heavy vehicle traffic and analysed the safety margins provided by AASHTO [11]. These authors analysed not only the increase in vertical deflections but also the bending moments at mid-span and at supports and concluded that, for good pavement surface conditions, American standard formulas [11] generally lead to conservative values of dynamic amplification. Therefore, the increase in vehicle circulation speed and the deterioration of the pavement surface stimulated researchers and structural engineers to initiate a continuous effort with the aim of improving the understanding of the dynamic phenomena, which has made it possible to establish new methodologies for the analysis and design of highway bridges.

In this regard, the following lines of studies aimed at highway bridge design stand out: (i) studies about the dynamic stress peaks with the vehicle–bridge interaction, which are obviously greater than the peaks obtained from the application of static or quasi-static loads, and (ii) studies on bridge vibrations, which, as is well known, cannot be excessive in order to minimize the effect of fatigue.

It should be highlighted that improvements in terms of innovation embodied in computers have made it possible to incentivize the application of the Finite Element Method (FEM) not only for the modelling of the structural components of bridges and vehicles but also for the modelling of pavement irregularities. In this way, the modelling process and simulations considering the vehicle–structure interaction and the progressive deterioration of the pavement make the dynamic responses more accurate and reliable so that the safety conditions of highway bridge design can be evaluated. However, despite the reality of the direct influence of the pavement surface condition on the structural dynamic response, some authors, including Skoglund and Leander [12], in recent studies have continued to perform fatigue analysis without considering the vehicle–structure interaction.

With these computational advances, using the refinement techniques present in Finite Element Method simulations, it becomes possible to precisely assess the fatigue phenomenon in structural elements of highway bridges, which is caused by the stochastic loading associated with vehicles crossing on the bridge deck pavement throughout their service life.

In this way, according to Fisher et al. [13], the structural details with welded joints are considered the most fragile points of steel and composite highway bridge designs, and almost 90% of all fatigue cracking cases are the result of out-of-plane distortion or other

unforeseen secondary stresses in fatigue-sensitive details. With regard to fatigue failure in welded joints, according to Klinger et al. [14], this process is subdivided into four stages: cyclic hardening, crack initiation, crack propagation and final failure. Although welded joints are evaluated in current design codes primarily using the Nominal Stress Method (NSM), in this study, for a more accurate stress definition, the Hot-spot Stress Method (HSM) was used.

Considering all these aspects, this investigation aimed to study, improve and implement a more advanced methodology for the evaluation of the fatigue phenomenon, considering the dynamic vehicle–structure interaction and the progressive deterioration of the pavement. Thus, a computational tool called VBI (Vehicle–Bridge Interaction) was developed in MATLAB [15] and comprises an interface with the finite element program ANSYS [16].

Thus, this research work aimed to develop an analysis methodology in order to assess not only the dynamic structural behaviour but also the fatigue behaviour of steel–concrete composite highway bridges, including the vehicle–structure interaction and the progressive pavement deterioration effect.

## 2. Methodology Framework

### 2.1. Vehicle–Bridge Dynamic Interaction Algorithm

Several research works [17–23] have been conducted, and it has been made evident that the effects due to the dynamic interaction between the vehicle's wheels and the irregular pavement surface can be much more important than those produced by the vehicle's smooth movement.

The dynamic vehicle–structure interaction and the pavement's progressive deterioration could be considered in the process of the fatigue damage evaluation carried out in the present research, taking into account the development of the computational tool VBI, which has an efficient algorithm for modelling and simulating this type of interaction. The automation of this analysis process was carried out with the help of complementary programs. The VBI tool code was developed in a MATLAB [15] environment in order to access the finite element simulation functionalities, and it interacts precisely with the ANSYS [16] program through command scripts.

It should be emphasized that the VBI application is based on another computational tool called Train-Bridge Interaction (TBI), developed by Ribeiro [24], which was designed to perform dynamic analyses considering the train–bridge interaction.

Therefore, the approach with the use of the VBI tool should be adopted to evaluate fatigue damage due to the precision and efficiency that it provides in the analysis process, considering the bridge's welded joint details [25] and taking into account the inclusion of the dynamic bridge–structure interaction and the pavement's progressive deterioration.

#### 2.1.1. Step 1: Modelling the Bridge and Vehicle

The first stage of the process consisted of developing the vehicle and bridge models based on the usual finite element modelling techniques present in the ANSYS [16] program. Thus, from these models, the mass and stiffness matrices of the bridge and the mass, stiffness and damping matrices of the vehicle were exported to the VBI tool in a MATLAB [15] environment in order to perform the dynamic analysis with the vehicle–bridge interaction.

#### 2.1.2. Step 2: Modelling the Most Critical Detail

In the second step, the critical detail model was also developed in the ANSYS [16] program, considering that the region with the greatest stress concentration effects is easily located and that the submodel boundaries are far enough away from the stress concentration in order to obtain accurate results from the calculations through the submodel.

It is important to note that the displacements in the global model of the bridge were applied to the investigated submodel as boundary conditions by performing an interpolation in the intercept region of the models (bridge model and critical detail submodel), taking

into account the positioning of both in relation to the coordinate system. The location of the submodel coordinates must be the same as those used in the global model.

The submodel was developed based on the Saint-Venant principle, which considers that the stress field of a region far away from the stress concentrations can be replaced for a set of equivalent displacements, since the distributions of stresses and strains undergo changes only in regions close to notches or with changes in geometry.

### 2.1.3. Step 3: Dynamic Analysis with Vehicle–Bridge Interaction

The third step consisted of performing a dynamic analysis with the vehicle–structure interaction, considering not only non-deterministic irregularities but also the pavement's progressive deterioration over time. This analysis was performed by using an algorithm originally developed by Ribeiro [24] and later expanded and modified by the authors of the present research [25]. The incorporated modifications concern the implementation of vehicles, non-deterministic road irregularities, the progressive deterioration of the pavement and the assessment of fatigue damage.

Concerning the numerical methodologies for the vehicle–structure interaction analysis, usually, the dynamic problem can be solved according to two different approaches: matrices of bridge and vehicle systems, coupled or uncoupled. In the coupled approach, the matrices of the bridge and vehicle systems are generated together, and, consequently, the dynamic equilibrium equations are solved. However, in the decoupled approach, which is the one used in the present paper, the dynamic equilibrium equations of the bridge and vehicles are generated separately, and the compatibility between the two structural systems is usually realized by using iterative methods, direct methods or contact algorithms.

It is important to emphasize that, in the decoupled approach adopted by the VBI tool, two different methodologies were used to solve the dynamic equilibrium equations: the modal superposition method and a direct integration method. Furthermore, it should be noted that the tool solves these dynamic equations separately and makes the two systems (vehicle and bridge) compatible through an iterative method.

In the case of the vehicle subsystem, the dynamic equations were solved by direct integration (Newmark's method), while, in the bridge subsystem, the dynamic equations were solved based on the modal superposition method. Thus, from a computational point of view, this approach is more efficient, as it allows a significant reduction in the bridge subsystem analysis time. In the Newmark formulation, the basic integration equations (Equations (1) and (2)) for the final velocity and displacement are expressed as follows [26]:

$$u(t + \Delta t) = u(t) + \Delta t \cdot \dot{u}(t) + \Delta t^2 \cdot (0.5 - \beta) \cdot \ddot{u}(t) + \Delta t^2 \cdot \beta \cdot \ddot{u}(t + \Delta t) \qquad (1)$$

$$\dot{u}(t + \Delta t) = \dot{u}(t) + [(1 - \gamma) \cdot \Delta t] \cdot \ddot{u}(t) + \Delta t \cdot \gamma \cdot \ddot{u}(t + \Delta t) \qquad (2)$$

where u, $\dot{u}$ and $\ddot{u}$ are the vectors of accelerations, velocities and displacements; t is time; $\Delta t$ is the time increment; and $\gamma$ and $\beta$ are the Newmark parameters ($\gamma = 1/2$ and $\beta = 1/4$ were adopted).

On the other hand, the modal superposition method is based on the transformation of the geometric coordinate problem into a modal coordinate problem with a system of independent linear equations. According to Chopra [27], the decoupling of the differential equations is performed by transforming the coordinates of real space (u) into coordinates in modal space ($y_n$), as shown in Equation (3):

$$M_n \cdot \ddot{y}_n(t) + C_n \cdot \dot{y}_n(t) + K_n \cdot y_n(t) = F_n(t) \qquad (3)$$

where $M_n$ represents the modal mass; $C_n$ is the modal damping; $K_n$ is the modal stiffness and $F_n$ is the modal force. Considering the orthogonality conditions between the mass and stiffness matrix and a Rayleigh damping matrix [27], Equations (4)–(7) are obtained:

$$M_n = \phi_n^T \cdot [M] \cdot \phi_n \qquad (4)$$

$$K_n = \phi_n^T \cdot [K] \cdot \phi_n \tag{5}$$

$$C_n = \phi_n^T \cdot [C] \cdot \phi_n \tag{6}$$

$$F_n = \phi_n^T \cdot \{F(t)\} \tag{7}$$

where n is the n-th vibration mode of the structure, and [M], [K] and [C] are the mass, stiffness and damping matrices, respectively. Therefore, after solving the equilibrium equations, the modal coordinates are calculated ($\ddot{y}_n(t)$, $\dot{y}_n(t)$ and $y_n(t)$) and, through the principle of the superposition of effects, the total responses in real space are equal to the sum of the contribution of the responses of the n modes in modal space, as shown in Equations (8)–(10):

$$u(t) = \sum_{n=1}^{N} \phi_n \cdot y_n(t) \tag{8}$$

$$\dot{u}(t) = \sum_{n=1}^{N} \phi_n \cdot \dot{y}_n(t) \tag{9}$$

$$\ddot{u}(t) = \sum_{n=1}^{N} \phi_n \cdot \ddot{y}_n(t) \tag{10}$$

Finally, with respect to the systems' compatibility, the bridge and the vehicle are considered two structural subsystems that are independently modelled. These structures (bridge and vehicle) are calculated simultaneously over time, and, at each time increment, an iterative process is performed in order to achieve the compatibility of the two structural subsystems in terms of dynamic interaction forces and displacements. Each time increment ($\Delta$t) involves the following operations performed at each iteration k [24]:

1. The mobile moving loads corresponding to the vehicle axles are applied to the bridge. Each moving load $F_p^k(t)$ is obtained from Equation (11):

$$F_p^k(t) = F_{sta} + F_{dyn}^{k-1}(t) \tag{11}$$

   where $F_{sta}$ is the static component of the interaction force, and $F_{dyn}^{k-1}(t)$ is the dynamic component of the interaction force relative to the previous iteration (equal to $F_{dyn}(\Delta t - t)$ for the first iteration).

2. At the same time, each contact point of the vehicle is subjected to the action of a supporting settlement, $u_v^k(t)$, corresponding to the displacement, $u_p^k(t)$, added to the eventual irregularity, r(t), at the point where the load is located. By solving the system of equations concerning the vehicle for each contact point, the support reactions $F_v^k(t)$ are obtained, which constitute the dynamic components of the interaction forces $F_{dyn}^k(t)$ to be applied to the bridge in the following iteration.

3. At the end of each iteration, a convergence criterion that takes into account the dynamic components of the interaction forces of the current and previous iteration is used for each moving load, as shown in Equation (12):

$$\frac{\left\| F_{dyn}^k(t) - F_{dyn}^{k-1}(t) \right\|}{\left\| F_{dyn}^{k-1}(t) \right\|} \tag{12}$$

If the resulting quotient is less than or equal to the given tolerance, it is considered that the two structural systems (bridge and vehicle) have been made compatible, moving on to the next instant (t + $\Delta$t); if it is not, a new iteration is performed. The process starts

by assuming that the dynamic components of the interaction force at the initial instant, $F_{dyn}(t = 0)$, are null. Table 1 illustrates the iterative methodology used in this investigation.

**Table 1.** Bridge–vehicle dynamic interaction: iterative methodology (adapted from Ribeiro [24]).

| | **Bridge** | **Vehicle** |
|---|---|---|
| Schema | | |
| Action | $F_p^k(t) = F_{sta} + F_{dyn}^{k-1}(t)$ | $u_v^k(t) = u_p^k(t) + r(t)$ |
| Result | $u_p^k(t)$ | $F_{dyn}^k(t) = F_v^k(t)$ |
| Convergence criterion | $\dfrac{\left\| F_{dyn}^k(t) - F_{dyn}^{k-1}(t) \right\|}{\left\| F_{dyn}^{k-1}(t) \right\|}$ | If $\leq$ tolerance $\rightarrow t + \Delta t$<br>If $>$ tolerance $\rightarrow k + 1$ |

2.1.4. Step 4: Modelling of the Road Surface Roughness

The road surface roughness was assumed as a zero-mean stationary Gaussian random process, based on studies carried out by Dodds and Robson [28], which can be generated through an inverse Fourier transformation, according to Equations (13) and (14):

$$r(x) = \sum_{i=1}^{N} \sqrt{2 \cdot \Delta\Omega \cdot G_d(\Omega_i)} \cdot \cos\left(2\pi \cdot \Omega_i \cdot x + \theta_i\right) \tag{13}$$

$$G_d(\Omega_i) = G_d(\Omega_0) \cdot \left[\frac{\Omega_i}{\Omega_0}\right]^{-2} \tag{14}$$

where $\theta_i$ is the random phase-angle, uniformly distributed from 0 to $2\pi$; $G_d(\Omega)$ is the Power Spectral Density (PSD) function ($cm^3$/cycle); $\Omega_i$ is the wave number (cycles/m); $\Omega_i$ is the spatial frequency of the pavement harmonic i (cycles/m); $\Omega_0$ is the discontinuity frequency of $\frac{1}{2}\pi$ (equal to 1 rad/m); $\Delta\Omega$ is the interval of discretization; and $G_d(\Omega_0)$ is the road roughness coefficient ($m^3$/cycle), also called RRC, whose values are presented in Table 2 [29] depending on the road class.

**Table 2.** Average values of $G_d(\Omega_0)$ for different levels of road quality (in $cm^3$) [29].

| Road Class | Road Quality Level | $G_d(\Omega_0)$: Lower | $G_d(\Omega_0)$: Mean | $G_d(\Omega_0)$: Upper |
|---|---|---|---|---|
| A | Excellent | - | 1 | 2 |
| B | Good | 2 | 4 | 8 |
| C | Average | 8 | 16 | 32 |
| D | Poor | 32 | 64 | 128 |
| E | Very poor | 128 | 256 | 512 |

In this context, with the purpose of contemplating the pavement's progressive deterioration, Paterson and Attoh-Okine [30] developed a model considering the International Roughness Index (IRI) with values at any time after starting the road surface service, which is calculated by Equations (15) and (16):

$$G_d(\Omega_0)_t = RRC_t = 6.1972 \cdot 10^{-9} \cdot \exp\left[IRI_t/0.42808\right] + 2 \cdot 10^{-6} \tag{15}$$

$$IRI_t = 1.04e^{nt} \cdot \left[IRI_0 + 263 \cdot (1 + SNC)^{-5} \cdot (CESAL)_t\right] \tag{16}$$

where $IRI_t$ is the value of IRI at time t; $IRI_0$ is the initial roughness value just after completing the construction and before opening it to traffic (set equal to 0.90 m/km [31]); t is the time in years; $\eta$ is the environmental coefficient, which depends on dry/wet, freezing/non-freezing conditions (set equal to 0.10 for bridges exposed to general environmental conditions); SNC

is the structural number, which is calculated from data on the strength and thickness of each layer in the pavement (set equal to 4); and (CESAL)t is the estimated traffic number in terms of the AASHTO [32] 80 kN cumulative equivalent single-axle load at time t, in millions, estimated using Equation (17):

$$(\text{CESAL})_t = f_d \cdot n_{tr}(t) \cdot F_{Ei} \cdot 10^{-6} \tag{17}$$

where $f_d$ is the design lane factor; $n_{tr}(t)$ is the cumulative number of vehicle passages for future year t; and $F_{Ei}$ is the load equivalence factor for axle category i.

Based on the *AASHTO Guide for Design of Pavement Structures* [32], the Equivalent Standard Axle Load (ESAL) of 80 kN, which is required to obtain (CESAL)$_t$, was calculated to be equal to 1.717 for the standard fatigue vehicle HL-93 (see Section 3.3) [32]. The adopted total number of vehicles per year is equal to 584,000, resulting from an average daily traffic volume of 1600 vehicles per lane crossing in one direction, taking into account the bridge's location (a place with medium vehicular traffic), as recommended by AASHTO [32]. In this context, (CESAL)$_t$ in the first year is equal to 1.0027 (1.717 · 584,000)/$10^6$ = 1.002728. It is important to note that this value changes if the number of vehicles per year is increased.

Therefore, by performing these necessary calculations, the pavement's progressive deterioration model was obtained. The road roughness classification was defined in accordance with ISO 8608 [29] (Table 3), and Figure 1 illustrates the RRC values calculated from Equation (15) on a logarithmic scale, aiming to obtain a better representation of the different roughness values.

**Table 3.** RRC values for road roughness classification, according to ISO 8608 [29].

| Road Roughness Classification | Ranges for RRCs |
|:---:|:---:|
| Very good | $2 \times 10^{-6}$ to $8 \times 10^{-6}$ |
| Good | $8 \times 10^{-6}$ to $32 \times 10^{-6}$ |
| Average | $32 \times 10^{-6}$ to $128 \times 10^{-6}$ |
| Poor | $128 \times 10^{-6}$ to $512 \times 10^{-6}$ |
| Very poor | $512 \times 10^{-6}$ to $2048 \times 10^{-6}$ |

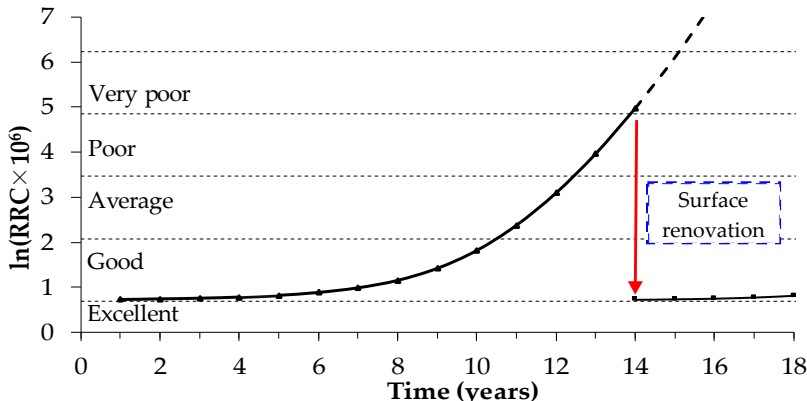

**Figure 1.** Increase rate of the deterioration of pavement road roughness in terms of $\ln(\text{RRC} \times 10^6)$ without traffic.

### 2.1.5. Step 5: Fatigue Strength Model

Several authors [33–35] in the field of engineering have discussed analyses related to the fatigue life of welded structures and classified the fatigue assessment into two parts: (1) the global analysis approach (nominal stress) and (2) local analysis, which is related to Hot-spot Stress Method. In this research work, the fatigue assessment was performed with the aid of the VBI tool, which applies the Palmgren–Miner rule based on the sum of

linear damage [36], expressed mathematically according to Equation (18), considering the hot-spot stress history obtained from the structural element or region of interest.

$$D = \frac{n_1}{N_1} + \frac{n_2}{N_2} + \frac{n_3}{N_3} + \ldots = \sum_{i=1}^{k} \frac{n_i}{N_i} \tag{18}$$

where D is the total damage; $n_i$ is the number of cycles at amplitude $\sigma_i$; and $N_i$ is the number of cycles to failure.

It is important to highlight that the influence of the load sequence effects and of the stress range cycles below the Constant Amplitude Fatigue Limit (CAFL) on the fatigue life is still under debate [37], and it is a vast field that is beyond the scope of the present thesis. For load histories that arise from stationary processes, such as traffic on bridges, the random occurrence of high and low stresses contributes to reducing the impact of the load sequence effects on the fatigue life [38]. Therefore, the direct consideration of load sequence effects on the fatigue life is disregarded in the current thesis, assuming that the adoption of a standard traffic model prescribed in structural codes for fatigue loading can reduce their impact since these models are based on a series of random stationary measurements of traffic loads on bridges.

The Hot-spot Stress Method (HSM) is the approach that determines the behaviour of a welded component considering the incorporation of the stress concentration effect induced by the weld geometry. This approach was initially developed for the analysis of fatigue in welded tubular joints of offshore structures [39,40]. Later, it was used for plate-like structures and is now replacing the stress-rated approach. In order to determine the hot-spot stress, the hot-spot or crack initiation point must be known in advance and must be accessible for evaluation.

After advancements in computational applications, researchers developed some finite element analysis methods to determine the hot-spot stress at the weld toe [41]. These methods are based on extrapolating the stress at different reference points on the plate surface (Figure 2) to the weld toe or linearizing the stress across the plate thickness. The fatigue assessment of structural details based on the Hot-spot Stress Method (HSM) should be considered extremely important, especially when the category of the detail that will be analysed does not exist in design codes. In this context, the IIW recommendations [33] are followed for a correct and more accurate assessment.

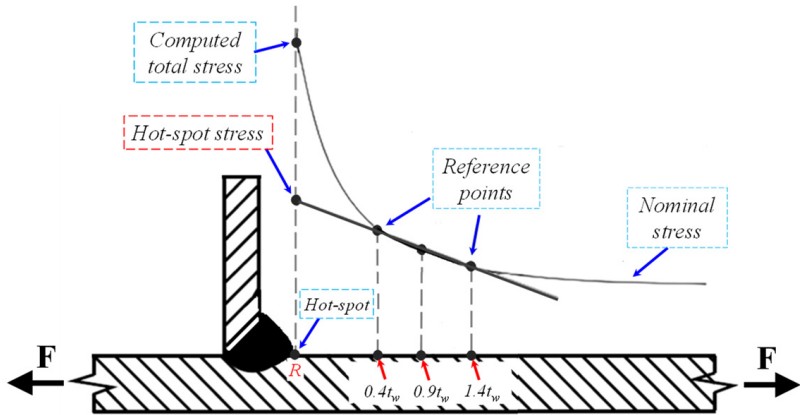

**Figure 2.** Typical representation of hot-spot stress extrapolation.

It is important to note that, in the literature, this extrapolated stress is often referred to as geometric stress, structural stress or hot-spot structural stress. Therefore, to simplify the understanding of this research work, all stress based on surface extrapolation is called hot-spot stress. As an illustration, it can be seen in Figure 2 that the hot-spot stress is obtained at point R based on the quadratic extrapolation of three reference points located at

distances of 0.4 $t_w$, 0.9 $t_w$ and 1.4 $t_w$ from the edge of the weld bead, according to Equation (19), where $t_w$ represents the plate thickness:

$$\sigma_{hs} = 2.52 \cdot \sigma_{0.4tw} - 2.24 \cdot \sigma_{0.9tw} + 0.72 \cdot \sigma_{1.4tw} \tag{19}$$

In fields with high bending stress, the IIW code [33] recommends performing quadratic extrapolation, while, in fields with low stress, it is possible to choose linear extrapolation simply by using two points at different distances.

The main routines of the VBI tool are illustrated in Figures 3 and 4 in flowcharts that present the proposed methodology for fatigue damage assessment.

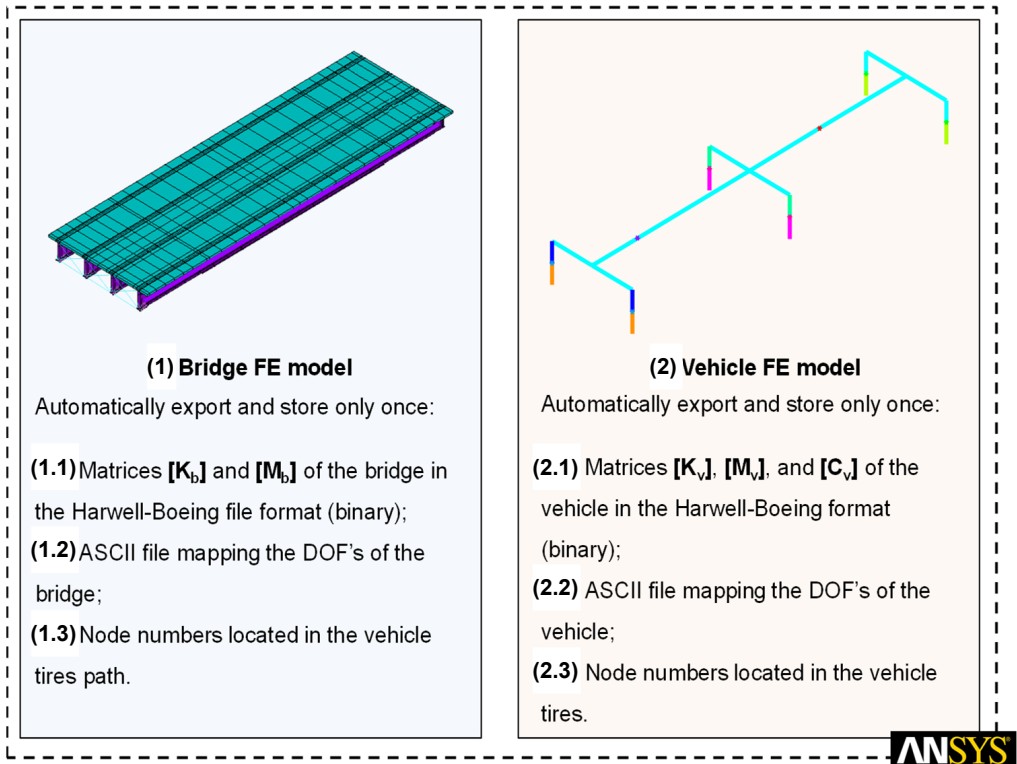

**(1) Bridge FE model**

Automatically export and store only once:

**(1.1)** Matrices **[K$_b$]** and **[M$_b$]** of the bridge in the Harwell-Boeing file format (binary);
**(1.2)** ASCII file mapping the DOF's of the bridge;
**(1.3)** Node numbers located in the vehicle tires path.

**(2) Vehicle FE model**

Automatically export and store only once:

**(2.1)** Matrices **[K$_v$]**, **[M$_v$]**, and **[C$_v$]** of the vehicle in the Harwell-Boeing format (binary);
**(2.2)** ASCII file mapping the DOF's of the vehicle;
**(2.3)** Node numbers located in the vehicle tires.

**Figure 3.** Flowchart of the proposed methodology: bridge FE model and vehicle FE model.

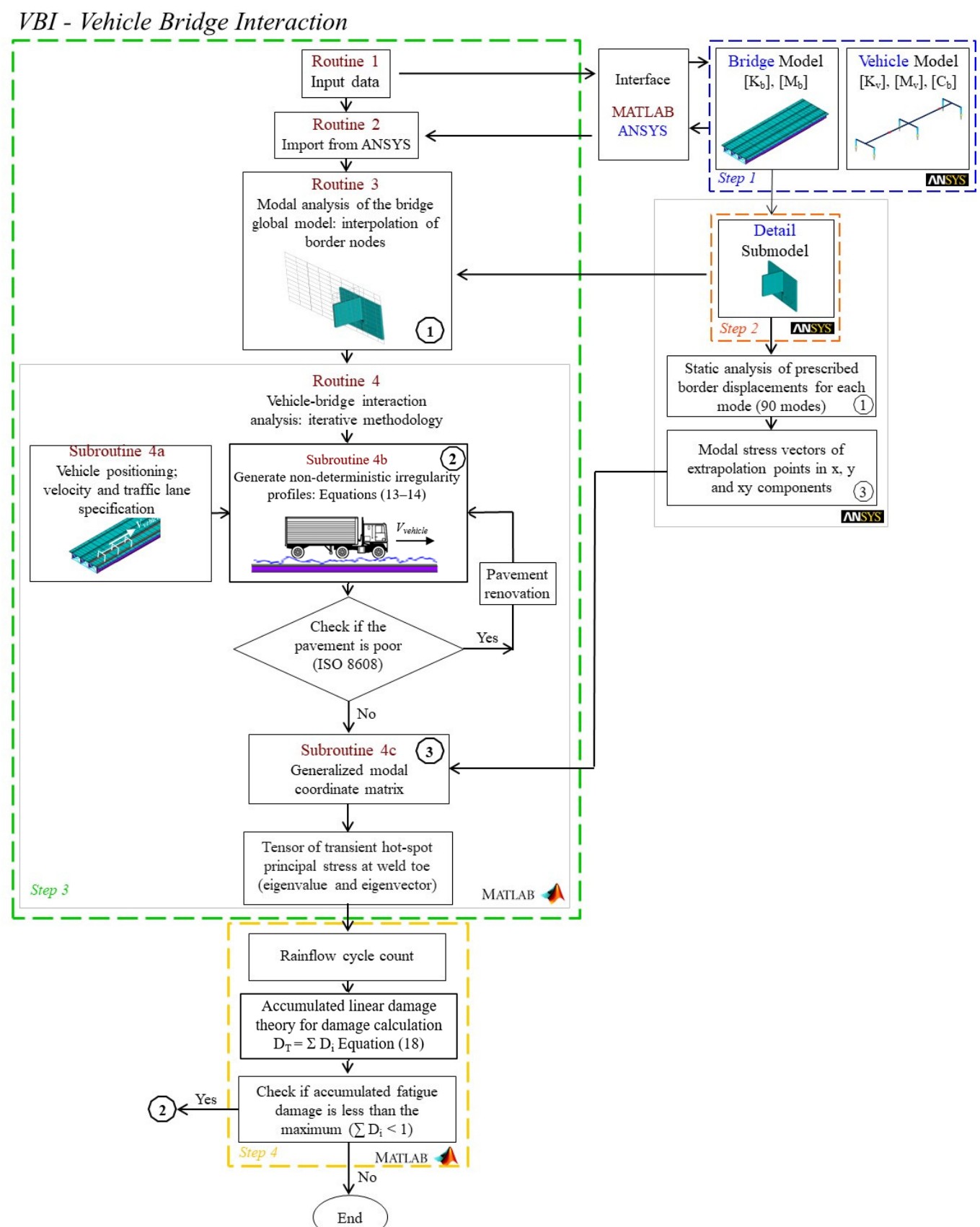

**Figure 4.** Flowchart of the proposed methodology: VBI tool.

## 3. Case Study

### 3.1. Background

In order to demonstrate the usefulness of the improved methodology proposed in the present research, a typical simply supported steel–concrete composite highway bridge with a straight axis spanning 13 m by 40 m was investigated as a case study (see Figure 5). The original design of this structure dates back to 2008 and was carried out by Pinho and Bellei [42] in accordance with AASHTO LRFD [43], and it was previously evaluated by Leitão et al. [44] in 2011 and Alencar et al. [31] in 2018. The structural system is constituted by four steel girders and a 0.225 m thick concrete slab, as shown in Figure 5 (see Table 4 for girder dimensions).

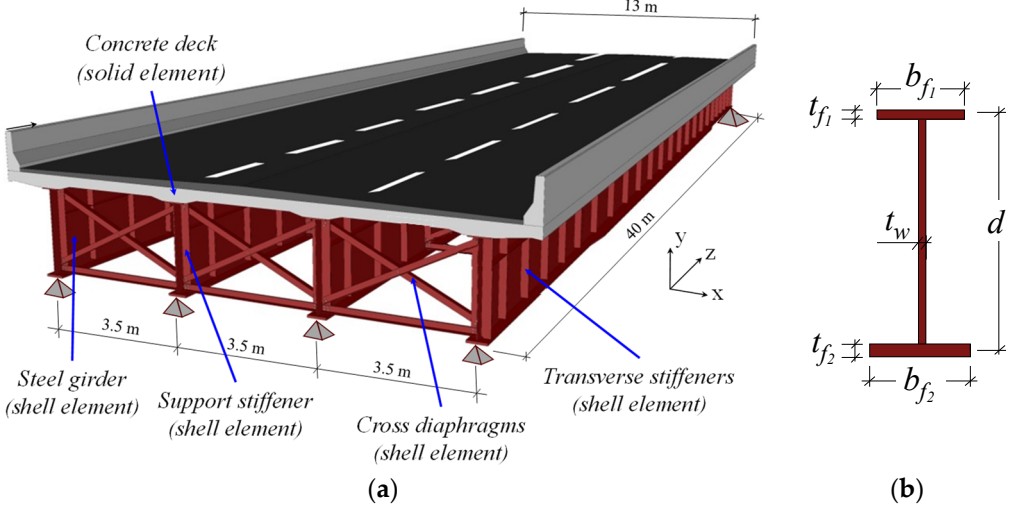

**Figure 5.** Steel–concrete composite highway bridge: (**a**) overall view; (**b**) steel girder profile (see Table 4 for dimensions).

**Table 4.** Geometrical characteristics of the steel girders (units in millimetres).

| Section Location | Height (d) | Top Flange Width ($b_{f1}$) | Top Flange Thickness ($t_{f1}$) | Bottom Flange Width ($b_{f2}$) | Bottom Flange Thickness ($t_{f2}$) | Web Thickness ($t_w$) |
|---|---|---|---|---|---|---|
| Support cross-section | 2000 | 450 | 25 | 450 | 50 | 9.5 |
| Span cross-section | 2000 | 500 | 25 | 670 | 50 | 9.5 |

It is important to note that two different cross-sections were adopted along the longitudinal composite beams: the support cross-section and the span cross-section (Figure 6). The steel sections considered are composed of wide welded flanges made with A588 steel with 350 MPa yield strength and 485 MPa ultimate tensile strength. A $2.05 \times 10^5$ MPa Young's modulus with a 0.3 Poisson's ratio and a material density of 7850 kg/m$^3$ were adopted for the steel girders. Regarding the concrete properties, the slab has a density of 2500 kg/m$^3$, Young's modulus of $3.05 \times 10^4$ MPa, Poisson's ratio of 0.2 and compressive strength of 25 MPa.

In order to prevent web buckling, steel plate stiffeners are welded along the steel girders with a spacing of 1880 mm in the span sections and 1200 mm in the support sections. The bridge structural system comprises cross diaphragms made of steel profiles with equal angles and a wall thickness of 10 mm; see Figure 7 and Table 5.

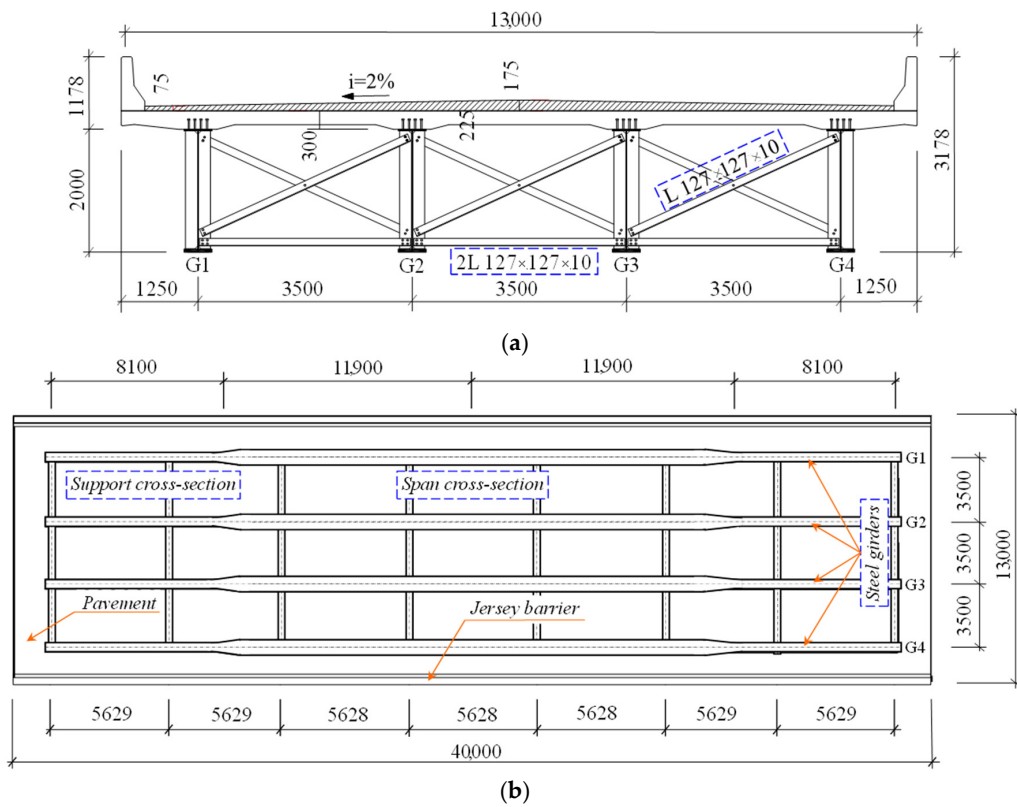

(**a**)

(**b**)

**Figure 6.** Geometry and dimensions: (**a**) section at support; (**b**) steel girder's top view (units in millimetres).

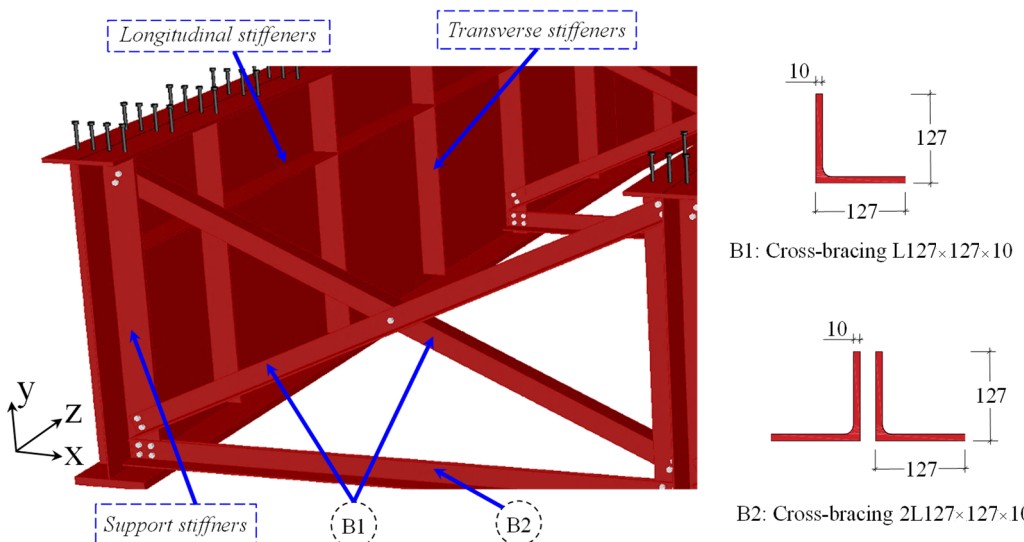

**Figure 7.** Cross-diaphragm sections and illustration of the web plate stiffeners.

**Table 5.** Geometrical characteristics of the web plate stiffeners (units in millimetres).

| Location | Height | Width | Thickness |
|---|---|---|---|
| Support stiffeners | 1925 | 200 | 22.0 |
| Transverse stiffeners | 1845 | 450 | 12.5 |
| Longitudinal stiffeners | --- | --- | 12.5 |

### 3.2. Numerical Model of the Bridge

As was mentioned before, the computational tool VBI used to evaluate the dynamic response with the vehicle–structure interaction was developed and implemented in the MATLAB [15] environment. Thus, this tool must initially import the structural matrices of the numerical models (bridge and vehicle) previously developed in finite element software. In the present research work, the numerical model of the typical composite highway bridge (Figure 5) was developed by adopting the usual mesh refinement techniques present in the Finite Element Method simulations implemented in the ANSYS program [16]. Shell finite elements were used to model the bridge girders' top and bottom flanges, the girder webs and the longitudinal and transverse vertical stiffeners; beam finite elements were used to model the transverse steel bracings; and solid elements were used to model the concrete slab.

Therefore, the structure was discretized into 4658 solid elements, 26,984 shell elements and 1136 beam elements, which resulted in a numerical model with a total of 40,832 nodes and 139,384 degrees of freedom. The strain compatibility between the solid elements (concrete slab) and the shell elements (steel plate girders) was guaranteed by coupling the corresponding degrees of freedom, simulating the composite bridge decks' full interaction. The damping ratio was assumed to be 0.005 ($\xi = 0.5\%$), as recommended by EN 1991-2 [45] for steel and steel–concrete composite bridges.

In sequence, Figure 8 illustrates the investigated highway bridge finite element model. The investigated steel–concrete composite bridge's natural frequencies and vibration modes were determined based on numerical methods of extraction (modal analysis) through a free vibration analysis using the ANSYS program [16]. The main associated global vibration modes of the bridge are shown in Figure 9.

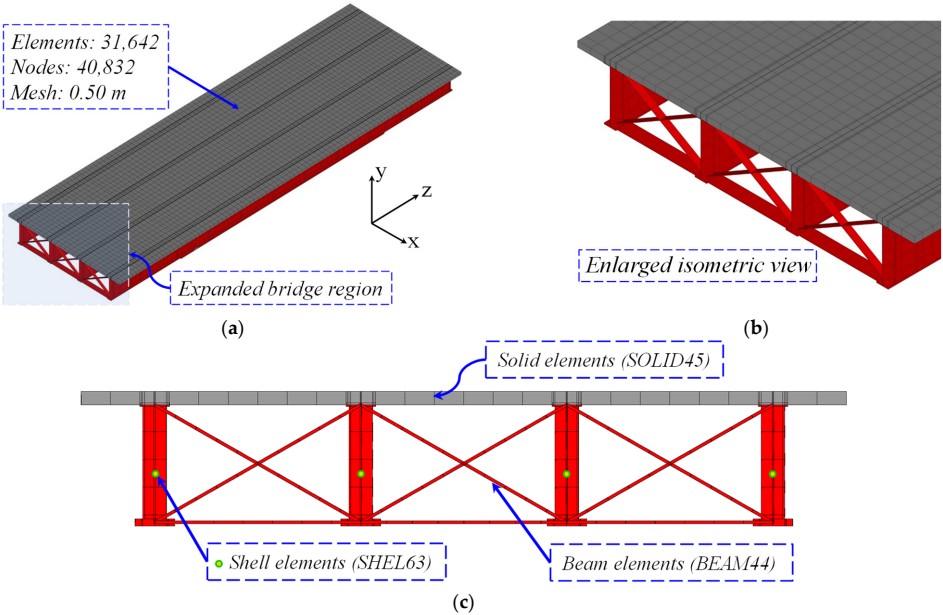

**Figure 8.** Finite element model: (**a**) overall view; (**b**) close-up view; (**c**) cross-section at supports.

### 3.3. HL-93 Vehicle

The vehicle considered in the present study was the HL-93 fatigue standard truck defined by AASHTO [43], which has three axles with 4.3 m spacing and axle loads of 35 kN in the front and 145 kN on the other two axles (intermediate and rear), as presented in Figure 10. The dynamic model of the vehicle is also shown in Figure 10b in a simplified schematic form, which is composed of two main sprung masses, representing the mechanical block and the bodies (rear), and three secondary masses, representing the vehicle axles located between the spring dampers, that simulate the behaviour of suspensions and tires. Springs and dampers are indicated with the letters k and c, respectively. The dynamic

properties (mass, damping and stiffness), including the tires and suspension systems, are listed in Table 6. These properties were determined based on studies conducted by Deng and Cai [4] and Montenegro et al. [46].

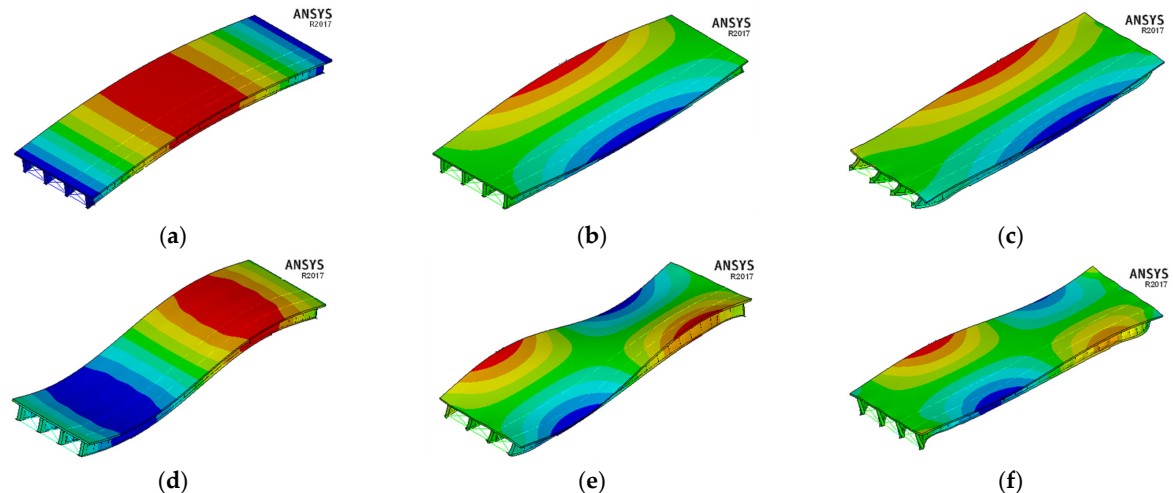

(a)                          (b)                          (c)

(d)                          (e)                          (f)

**Figure 9.** Main global vibration modes of the investigated bridge: (**a**) $f_{01}$ = 2.54 Hz; (**b**) $f_{02}$ = 3.13 Hz; (**c**) $f_{03}$ = 5.34 Hz; (**d**) $f_{04}$ = 8.25 Hz; (**e**) $f_{05}$ = 9.27 Hz; (**f**) $f_{06}$ = 10.14 Hz.

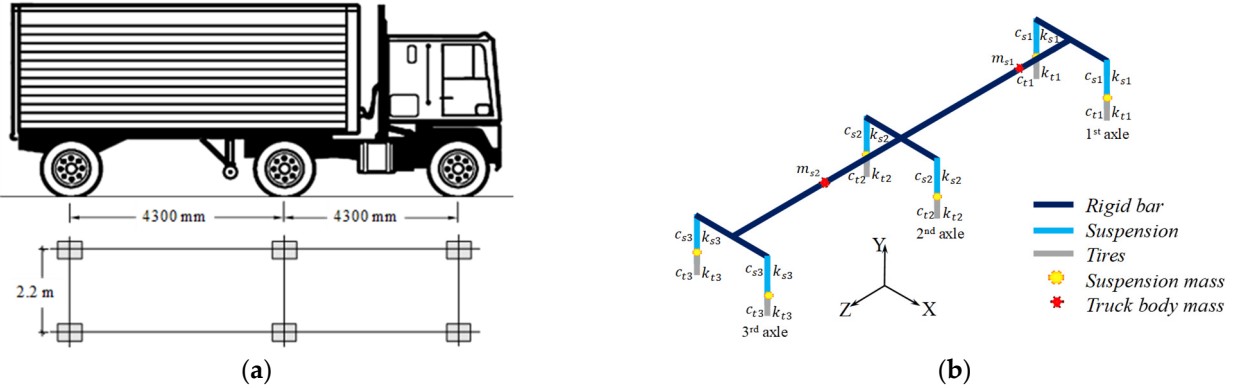

(a)                                              (b)

**Figure 10.** Vehicle HL-93: (**a**) fatigue standard vehicle (adapted from [43]); (**b**) schematic dynamic model (adapted from [16]).

**Table 6.** Mechanical and geometric properties: HL-93 vehicle model [4,46].

| Parameter | Value | Units | Parameter | Value | Units |
|---|---|---|---|---|---|
| Front truck body mass ($m_{s1}$) | 2612 | kg | 2nd axle tire spring stiffness ($k_{t2}$) | 3503 | N/m |
| Pitching—front block rotational inertia ($I_{x,cf}$) | 8544 | kg.m$^2$ | 2nd axle tire damping ($c_{t2}$) | 2000 | N.s/m |
| Rolling—front block rotational inertia ($I_{z,cf}$) | 2022 | kg.m$^2$ | 3rd axle suspension mass ($m_{a3}$) | 653 | kg |
| Rear truck body mass ($m_{s2}$) | 28,077 | kg | Rolling—rear axle rotational inertia ($I_{z,ar}$) | 600 | kg.m$^2$ |
| Pitching—rear block rotational inertia ($I_{x,cr}$) | 181,216 | kg.m$^2$ | 3rd axle suspension spring stiffness ($k_{s3}$) | 1,969,034 | N/m |
| Rolling—rear block rotational inertia ($I_{z,cr}$) | 33,153 | kg.m$^2$ | 3rd axle damping ($c_{a3}$) | 7182 | N.s/m |
| 1st axle suspension mass ($m_{a1}$) | 490 | kg | 3rd axle tire spring stiffness ($k_{t3}$) | 3,507,429 | N/m |
| 1st axle suspension spring stiffness ($k_{s1}$) | 242,604 | N/m | 3rd axle tire damping ($c_{t3}$) | 2000 | N.s/m |
| 1st axle damping ($c_{a1}$) | 2190 | N.s/m | Dist. from 1st axle to front block ($L_1$) | 1240 | mm |
| 1st axle tire spring stiffness ($k_{t1}$) | 875,082 | N/m | Dist. from front block to 2nd axle ($L_2$) | 3060 | mm |
| 1st axle tire damping ($c_{t1}$) | 2000 | N.s/m | Dist. from 2nd axle to rear block ($L_3$) | 1925 | mm |
| 2nd axle suspension mass ($m_{a2}$) | 808 | kg | Dist. from rear block to 3rd axle ($L_4$) | 2375 | mm |
| Rolling—central axis rotation inertia ($I_{z,am}$) | 600 | kg.m$^2$ | Dist. from front block to rear block connection ($L_5$) | 2673 | mm |
| 2nd axle suspension spring stiffness ($k_{s2}$) | 1,903,172 | N/m | Dist. from connection to rear block | 2312 | mm |
| 2nd axle damping ($c_{a2}$) | 7882 | N.s/m | Transverse distance between tires ($B_t$) | 2200 | mm |

As was mentioned before, the vehicle model was developed in ANSYS [16]. Thus, the two suspended masses are connected by rigid-beam finite elements, while the suspensions are simulated by the damper springs (see Figure 10b). The vehicle's associated global vibration modes and natural frequencies are shown in Figure 11.

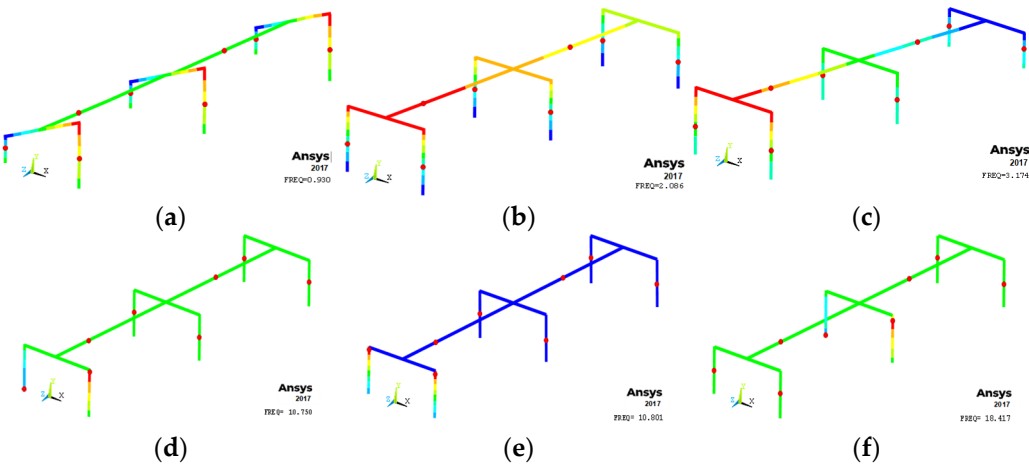

**Figure 11.** Global vibration modes of HL-93: (**a**) $f_{01}$ = 0.93 Hz; (**b**) $f_{02}$ = 2.09 Hz; (**c**) $f_{03}$ = 3.17 Hz; (**d**) $f_{04}$ = 10.75 Hz; (**e**) $f_{05}$ = 10.80 Hz; (**f**) $f_{06}$ = 18.42 Hz.

It is important to emphasize that the vehicle velocity distribution per year used in this work was developed by Rossigali [47] (see Table 7) and is based on a normal distribution (Gaussian distribution).

**Table 7.** Adopted vehicle velocity distribution per year [47].

| Velocity | 40 km/h | 60 km/h | 80 km/h | 100 km/h | 120 km/h |
|---|---|---|---|---|---|
| Distribution | 1.81% | 21.92% | 63.94% | 12.06% | 0.27% |

## 4. Fatigue Assessment Methodology

*Finite Element Modelling Methodology and Assessment of Fatigue-Prone Detail*

The identification of the region most prone to fatigue, which is subject to the highest stress amplitudes, is a crucial step for fatigue assessment in steel and composite bridges. Since it is very difficult to determine the nominal stress in the web region close to the transverse stiffener due to the high secondary stress involved, it is essential to use the Hot-spot Stress Method in order to provide a more accurate assessment of fatigue damage. The welded structural component investigated in this study consisted of a FAT90 [33] category detail according to the S-N design approach.

The most critical region in the investigated bridge is clearly located in the middle of the span on the most extreme girder, G4 (Figure 12c). In this region, the most critical transverse stiffener weld ends are located, where eccentric traffic loading is responsible for the in-plane and out-of-plane bending of the web. Therefore, in view of the above, it is justified to carry out the submodelling of the critical detail that consists of the transverse stiffener welded to the web (Figure 13a).

In the submodel, solid elements (SOLID186) were used with functions in a quadratic form according to IIW guidelines [33]. Regarding the mesh, fine mesh and quadratic extrapolation with three reference points were used. The fillet welds were modelled, but the structural imperfections/misalignments were not considered, since all stress concentrations due to the former are already included in the hot-spot stress determination, and any allowance for misalignment is covered in the hot-spot SN curves within a limit of 5% stress magnification [31,34]. The use of the submodelling technique allows one not only to reduce computational costs but also to obtain reliable estimates of the gradient stress at the weld toe.

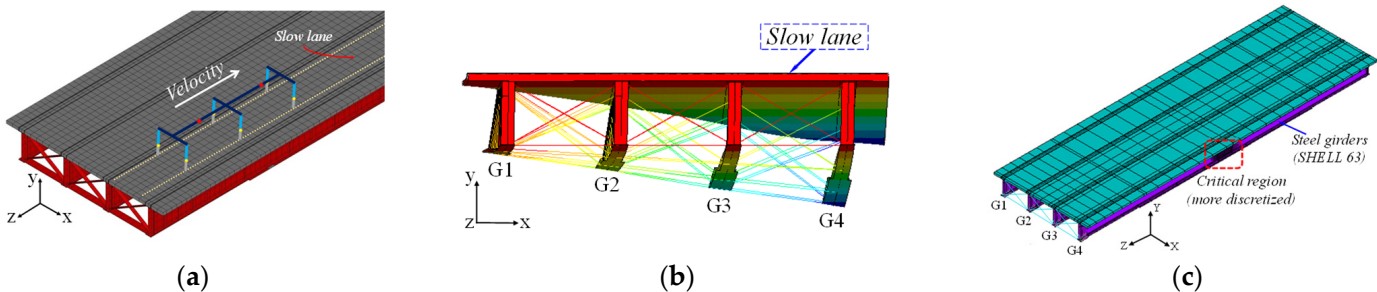

**Figure 12.** Displacement in the mid-span: (**a**) vehicle in the slow lane; (**b**) frontal view; (**c**) isometric view with the critical region.

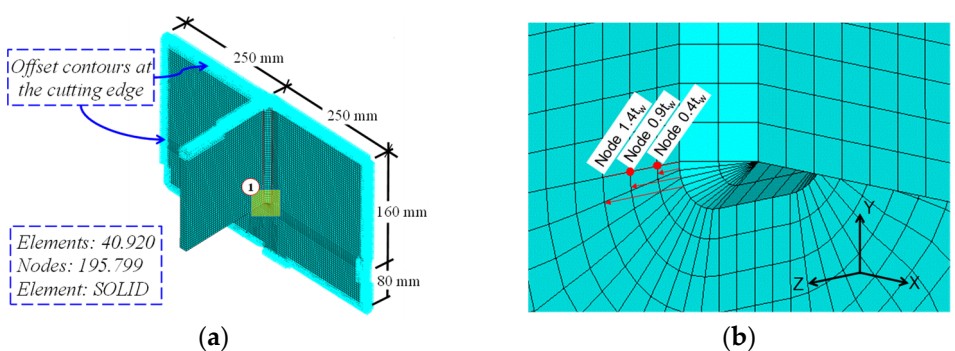

**Figure 13.** Submodel: (**a**) structural detail of the local model in finite elements; (**b**) region for fatigue assessment induced by distortion in the submodel.

As previously described, the stress history components were calculated from the superposition solution, taking into account the different levels of surface roughness and vehicle velocities. The main stress histories, obtained based on the integration of MATLAB [15] and ANSYS [16] (Figure 4), were used to calculate the fatigue damage considering IIW recommendations [33].

The critical hot-spot points along the weld of the web-gap detail in the investigated submodel are shown in Figure 14. For a better understanding, the stiffener welded to the web was separated into two details: Detail A (closest to the mid-span) and Detail B (farther away from the mid-span). Moreover, the hot-spot extrapolation paths around the corners of the investigated weld are also shown at 22 nodes, making 11 sets with three extrapolation points at each of the details (Details A and B).

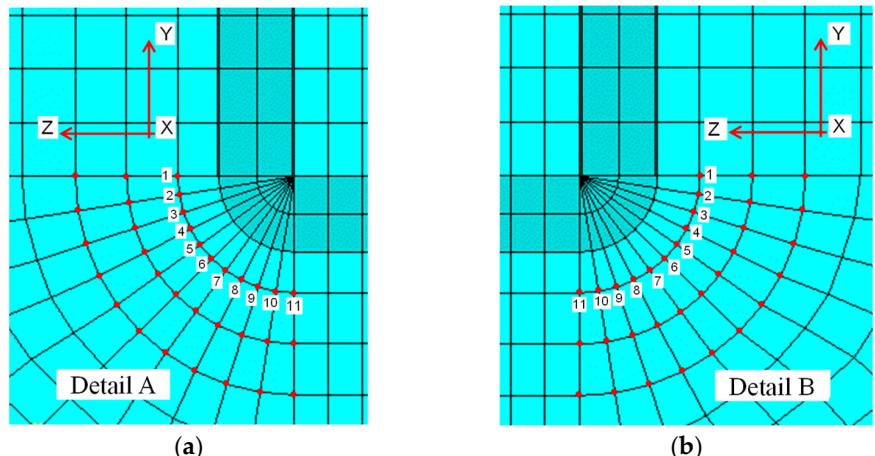

**Figure 14.** Front view of the submodel: hot-spot extrapolation paths around the corners: (**a**) Detail A (closest to the mid-span); (**b**) Detail B (farther away from the mid-span).

The hot-spot stress was obtained with three extrapolation points at each of the details (Details A and B). The methodology used to apply the Hot-spot Stress Method was previously validated by the authors [48] based on experimental results obtained by Fisher et al. [13] for a structural detail and loading very similar to those of the present investigation.

## 5. Comparative Framework of the Presented Methodology with Advancements in Relation to Previous Works

In the present section, the authors present a comparative framework of the developed methodology (Table 8) with advancements in relation to previous works. The methodology for the dynamic analysis and fatigue assessment of steel–concrete bridges was developed by the Structural Dynamics research group of the State University of Rio de Janeiro (UERJ) in 2009 in collaboration with the University of Brasília (UnB) and the University of Porto (Portugal).

**Table 8.** Advancements in fatigue analysis methodologies for composite highway bridges.

| Leitão et al. (2011) in *Journal of Constructional Steel Research* [44] | Alencar et al. (2018) in *Engineering Structures* [31] | Silva et al. (2023) in *Metals* (Present Work) |
|---|---|---|
| Main contributions and advancements | Main contributions and advancements | Main contributions and advancements |
| • Use of a detailed 3D finite element model for the bridge.<br>• The first time that a fatigue analysis was carried out for a steel and composite (steel–concrete) bridge using a 3D model and the stresses from a dynamic bridge–vehicle interaction analysis were obtained while considering the irregularities of the pavement.<br>• Performs the full integration of the dynamic equilibrium equations of the bridge–vehicle system with the Newmark numerical method for the 2D model and then imports the dynamic contact forces into the ANSYS environment with a detailed 3D model for the bridge; also solves a dynamic moving load problem with the full integration of dynamic equilibrium equations. | • Use of a detailed 3D finite element model for the structural detail (multiscale and submodelling approach).<br>• Use of more precise hot-spot S-N curves, with clear computation of the stress field around weld notches.<br>• Performs the full integration of the dynamic equilibrium equations of the bridge+vehicle system with the Newmark numerical method only for the 2D model.<br>• Solves the problem of dynamic moving loads across the 3D bridge with the mode superposition method (much less computer-intensive).<br>• With the aid of the mode superposition method, it allows the study of the influence of local vibration modes on the amplification of local stresses in certain details (global and local resonance).<br>• Inspired by the works of C.S. Cai [4,10,19], this work included the implementation of a function for the deterioration of the pavement, thus allowing the use of multiple levels of roughness for the degraded pavement throughout the entire life-cycle analysis. | • Supports any vehicle model modelled with the aid of finite element software (no need to derive the equations of motion of each vehicle by hand).<br>• Can be used for any kind of straight bridge with varied design solutions for which the structural behaviour can be predicted with linear dynamics (in other words, it is not limited to bridges that can be well approximated by 2D models). Vehicles can be positioned transversely in any position on the road.<br>• Fatigue strength (hot-spot S-N curve) now supported by the reanalysis of experimental data for distortion-induced fatigue-prone details (web-gaps) from a previous work [48]—a bilinear S-N curve FAT90 according to IIW.<br>• Pavement irregularities of both tire paths can now be modelled differently.<br>• Maintains all the advancements from Alencar et al. [31].<br>• The methodology is able to model a complicated traffic flow, with different types of vehicles spanning different lengths. |
| Drawbacks | Drawbacks | Future challenges |
| • Use of less accurate nominal S-N curves.<br>• Poor stress analysis due to the multiscale problem in fatigue analysis. In this sense, stress could be considered accurate only for details with a clear and uniform stress field, far from notches and intricated geometries.<br>• Use of only one level of roughness throughout the entire life-cycle.<br>• Limited to bridge modes that can be approximated with moderate accuracy by simplified 2D model, thus neglecting torsion effects induced by vehicles.<br>• Vehicle equations of motion and matrices need to be derived by hand (dynamic equilibrium through D'Alembert principle).<br>• Imports the contact forces from the simplified 2D model for the ANSYS environment (3D bridge) and solves the dynamic equilibrium equations of the bridge by performing full integration.<br>• Pavement irregularities of both tire paths are modelled as equal. | • Limited to bridge models that can be approximated with moderate accuracy by simplified 2D model, thus neglecting torsion effects induced by vehicles positioned transversely out of the bridge axis.<br>• Vehicle equations of motion and matrices need to be derived by hand calculations (dynamic equilibrium through D'Alembert principle)<br>• Roughness of both tire paths is modelled as equal. | • To allow the modelling and use of curved and skewed bridges.<br>• To implement more types of vehicles and integrate a Monte Carlo simulation.<br>• To adopt more advanced pavement deterioration models that are less dependent on empirical data. |

The framework also discusses the drawbacks of each work, along with future challenges and next steps. First published by Leitão et al. in 2011 [44], the method was improved by Alencar et al. in 2018 [31] with more advanced fatigue stress definitions (local approach) and improved dynamic analysis with the mode superposition method, and it is now further improved in the present work of Silva et al. in 2023, with the ability to model any type of vehicle and to use it for bridges with more varied forms than the classical straight-axis bridge type.

## 6. Results and Discussion

### 6.1. Dynamic Analysis with Vehicle–Bridge Interaction

Comparison of Numerical Response in Terms of Displacements

This section presents the results of the comparison of displacements obtained based on the dynamic analysis with the vehicle–structure interaction at mid-span considering two different roughness conditions, good and poor, taking into account a single three-axle vehicle crossing at ν = 40 km/h (Figures 15 and 16).

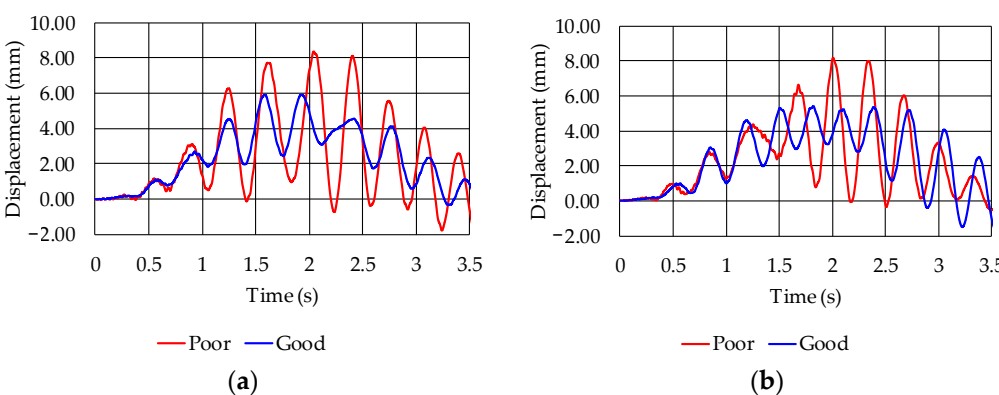

**Figure 15.** Vertical displacements at mid-span (set 1): (**a**) Detail A; (**b**) Detail B.

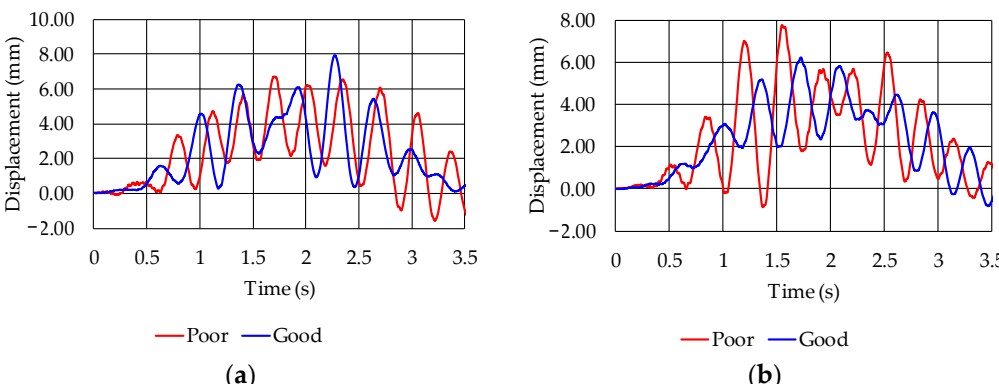

**Figure 16.** Vertical displacements at mid-span (set 11): (**a**) Detail A; (**b**) Detail B.

As expected, the displacements obtained in a more deteriorated pavement condition (poor) were greater than those obtained in a superior condition (good). Moreover, it is possible to observe that, with respect to displacements, both directions analysed (sets 1 and 11) present qualitatively similar results for Detail A and Detail B.

### 6.2. Dynamic Analysis with Interaction: Identification of the Critical Hot-Spot

In order to identify the most critical direction, ν = 40 km/h and $RRC_{renewal\ limit}$ of $4 \times 10^{-6}$ and $64 \times 10^{-6}$ were considered (Figure 17). For a better understanding, the stiffener welded to the web was separated into two details: Detail A (closest to the mid-span) and Detail B (farther away from the mid-span); see Figure 14. The results of the hot-spot stress at the various points along the weld in the web-gap detail are presented in

Figures 18 and 19. It should be noted that the results illustrated below refer to the worst case of increased traffic ($\alpha = 5\%$).

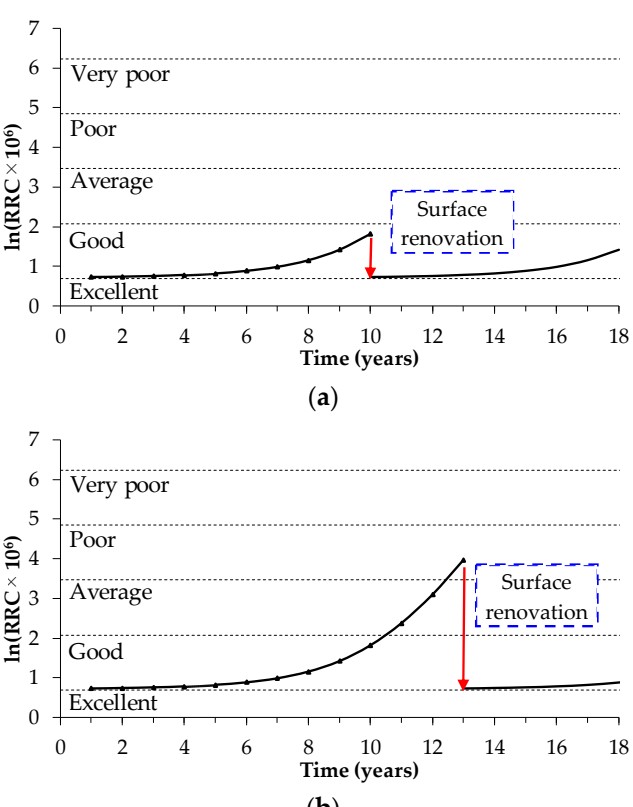

(a)

(b)

**Figure 17.** Surface renovation examples: (**a**) $RRC_{renewal\ limit} = 4 \times 10^{-6}$; (**b**) $RRC_{renewal\ limit} = 64 \times 10^{-6}$.

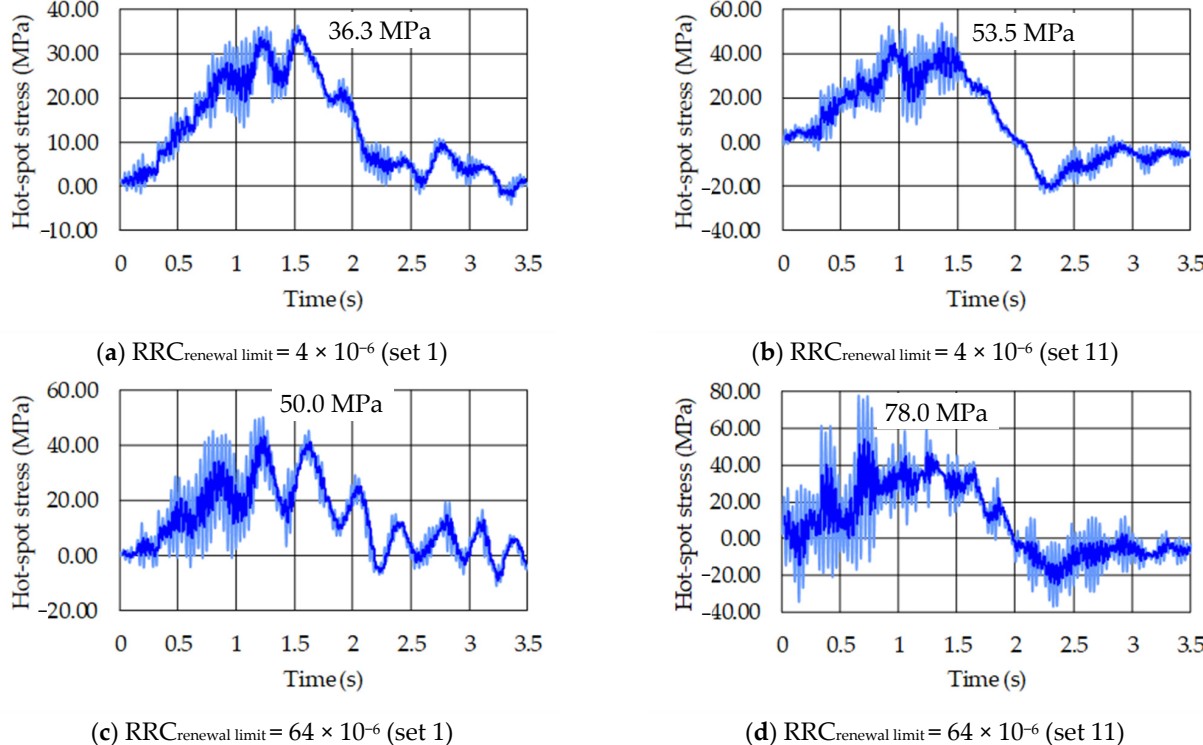

(**a**) $RRC_{renewal\ limit} = 4 \times 10^{-6}$ (set 1)

(**b**) $RRC_{renewal\ limit} = 4 \times 10^{-6}$ (set 11)

(**c**) $RRC_{renewal\ limit} = 64 \times 10^{-6}$ (set 1)

(**d**) $RRC_{renewal\ limit} = 64 \times 10^{-6}$ (set 11)

**Figure 18.** Hot-spot stress: submodel web-gap (Detail A).

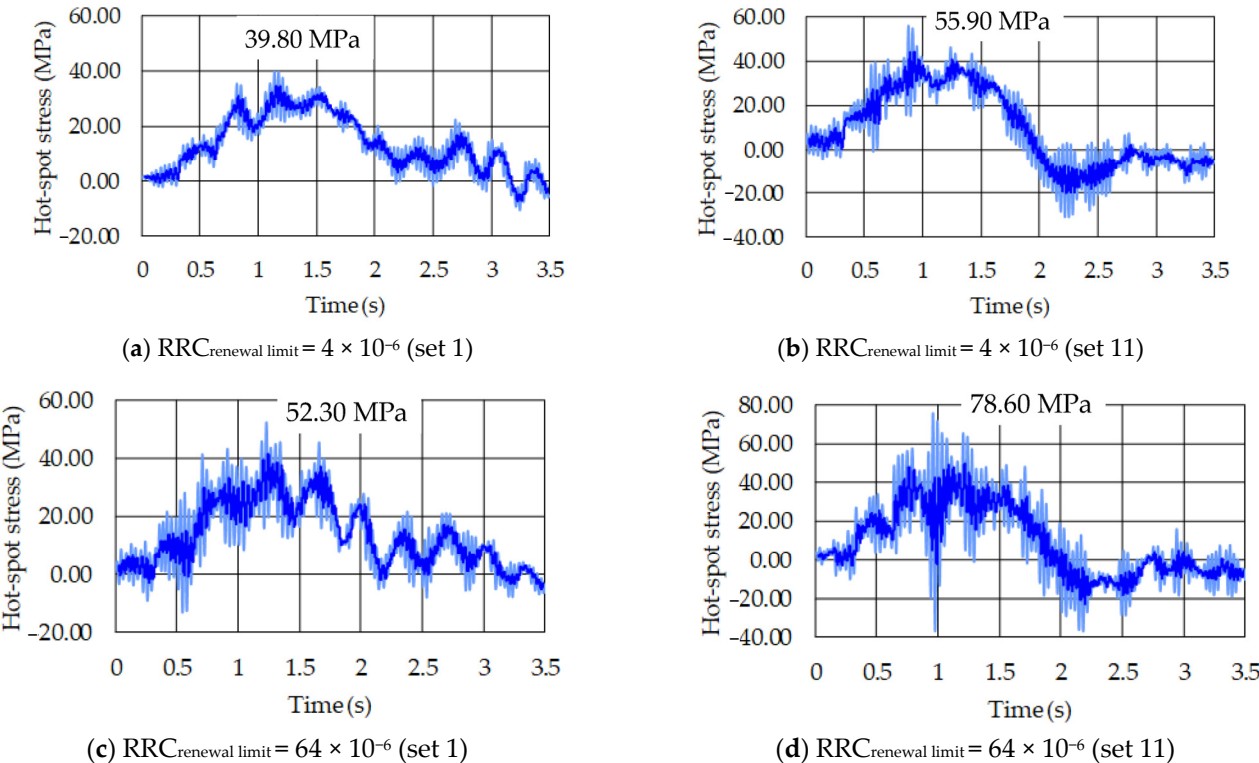

(a) RRC$_\text{renewal limit}$ = 4 × 10$^{-6}$ (set 1)

(b) RRC$_\text{renewal limit}$ = 4 × 10$^{-6}$ (set 11)

(c) RRC$_\text{renewal limit}$ = 64 × 10$^{-6}$ (set 1)

(d) RRC$_\text{renewal limit}$ = 64 × 10$^{-6}$ (set 11)

**Figure 19.** Hot-spot stress: submodel web-gap (Detail B).

Based on the results presented (Figures 18 and 19), it is observed that Detail B is the most critical in both directions analysed (sets 1 and 11) and in both situations of surface renovation. Regarding the analysed directions (sets 1 and 11), it is observed that direction 11 is the most critical for both details investigated; see Figures 18b,d and 19b,d. With respect to the renewal class, as expected, when the renewal limit is inferior ($4 \times 10^{-6}$), the stresses are lower, since the renewal of the pavement happens when the pavement is still in good condition; see Figures 18a,b and 19a,b.

Thus, in relation to these values of out-of-plane hot-spot stresses obtained in the weld in the most critical component due to the crossover of the standard fatigue vehicle (v = 40 km/h) considering the worst direction (set 11), the calculated results were 53.5 MPa and 78 MPa for Detail A, respectively, for RRC$_\text{renewal limit}$ values of $4 \times 10^{-6}$ and $64 \times 10^{-6}$. On the other hand, for Detail B, still considering the worst direction (set 11), the calculated results were 55.90 MPa and 78.60 MPa for RRC$_\text{renewal limit}$ values of $4 \times 10^{-6}$ and $64 \times 10^{-6}$, respectively.

*6.3. Fatigue Life Estimation: Damage Evolution with Pavement Deterioration*

As discussed before, the fatigue life estimates were performed based on the Palmgren–Miner rule, considering the fatigue strength characteristic for a constant amplitude of $2 \times 10^6$ cycles. In this context, the history of hot-spot stress obtained in the most critical component was considered, which was located at the weld toe between the stiffener and the web in the detail of the submodel. Tables 9 and 10 present the accumulated fatigue damage, expressed in years, for both investigated details as a function of the RRC$_\text{renewal limit}$ variation with traffic increase rates of 0%, 3% and 5%.

Regarding the hot-spot stress assessment, initially, a linear extrapolation was performed, followed by a quadratic extrapolation for the evaluated structural detail in the position closest to the mid-span. Significant differences in the damage calculations and, consequently, in the useful lifetime were observed, because there was a representative variation in the stress variation amplitudes. In this sense, the computational time also presented a relevant difference, and quadratic extrapolation was selected for the assessments

performed in the present work. The computational time can be exponentially reduced because of the submodelling and the interpolation of the displacement field between the global model and the submodel.

**Table 9.** Detail A: fatigue life for different RRC renewal limits and traffic increase rates.

| Detail A: Definite direction = 1 | | |
|---|---|---|
| Traffic increase rate | $RRC_{renewal\ limit} = 4 \times 10^{-6}$ (Road quality level: good) | $RRC_{renewal\ limit} = 64 \times 10^{-6}$ (Road quality level: poor) |
| $\alpha = 0\%$ | >100 years | 94 years |
| $\alpha = 3\%$ | 98 years | 25 years |
| $\alpha = 5\%$ | 70 years | 24 years |
| Detail A: Definite direction = 11 | | |
| Traffic increase rate | $RRC_{renewal\ limit} = 4 \times 10^{-6}$ (Road quality level: good) | $RRC_{renewal\ limit} = 64 \times 10^{-6}$ (Road quality level: poor) |
| $\alpha = 0\%$ | 40 years | 12 years |
| $\alpha = 3\%$ | 30 years | 11 years |
| $\alpha = 5\%$ | 24 years | 11 years |

**Table 10.** Detail B: fatigue life for different RRC renewal limits and traffic increase rates.

| Detail B: Definite direction = 1 | | |
|---|---|---|
| Traffic increase rate | RRC renewal limit = $4 \times 10^{-6}$ (Road quality level: good) | RRC renewal limit = $64 \times 10^{-6}$ (Road quality level: poor) |
| $\alpha = 0\%$ | >100 years | 61 years |
| $\alpha = 3\%$ | 94 years | 37 years |
| $\alpha = 5\%$ | 65 years | 24 years |
| Detail B: Definite direction = 11 | | |
| Traffic increase rate | RRC renewal limit = $4 \times 10^{-6}$ (Road quality level: good) | RRC renewal limit = $64 \times 10^{-6}$ (Road quality level: poor) |
| $\alpha = 0\%$ | 44 years | 12 years |
| $\alpha = 3\%$ | 31 years | 11 years |
| $\alpha = 5\%$ | 25 years | 11 years |

It should be noted that, as expected, the scenarios of the traffic increase directly influence the response of the structure. The higher the growth rate, the greater the displacements and stresses and, consequently, the shorter the estimated fatigue life. Furthermore, it can be observed that renewal limit changes significantly alter the useful lifetime. If the renovation is performed in an already-deteriorated condition (poor), the useful life is greatly reduced.

## 7. Conclusions

In this research work, a numerical methodology of dynamic analysis is described for the fatigue assessment of steel and steel–concrete composite highway bridges according to IIW and AASHTO recommendations. The conducted investigation involved extensive dynamic analyses that required the implementation of the VBI computational tool, which considers not only the vehicle–bridge interaction but also the pavement's progressive deterioration. The following conclusions can be drawn from the results presented in this work:

- The pavement surface condition directly influences the analysis responses. A more deteriorated condition induces higher values of displacement and stress.
- The position of the detail also influences the response of the structure. Based on the results presented, it is observed that Detail B (farther away from the mid-span) is the most critical in both directions analysed and in both situations of surface renovation.

- The hot-spot extrapolation paths around the corners are also parameters that influence the structural response. Considering Detail B, the percentage difference between the hot-spot stresses in the two directions analysed is 40.45% for $RRC_{renewal\ limit}$ values of $4 \times 10^{-6}$ and 50.28% for an $RRC_{renewal\ limit}$ of $64 \times 10^{-6}$. Thus, it can be observed that direction 11 is the worst case for both analysed details.

- The scenarios of the traffic increase also directly influence the response of the structure. The higher the growth rate, the greater the displacements and stresses and, consequently, the shorter the estimated fatigue life.

- The renewal limit changes significantly alter the useful lifetime. If the renovation is performed in an already-deteriorated condition (poor), the useful life is greatly reduced. Considering Detail B and set 11 (worst case), the lifetime of the detail increased considerably, by about 266%, 181.11% and 127.27%, respectively, for traffic increase rates of 0%, 3% and 5% when the pavement renovation was carried out when the surface was still in good condition ($RRC_{renewal\ limit} = 4 \times 10^{-6}$).

- Thus, the importance of more effective maintenance of the pavement is highlighted, as it will ensure a longer useful lifetime for the structure. Therefore, it is understood that if pavement renovation is carried out earlier, when the pavement condition is not so deteriorated, the useful lifetime can be extended, reaching the time required by the design standards.

**Author Contributions:** Conceptualization, A.C.S.d.S., G.S.A. and J.G.S.d.S.; methodology, A.C.S.d.S., G.S.A. and J.G.S.d.S.; software, A.C.S.d.S., G.S.A. and J.G.S.d.S.; validation, A.C.S.d.S., G.S.A. and J.G.S.d.S.; formal analysis, A.C.S.d.S., G.S.A. and J.G.S.d.S.; investigation, A.C.S.d.S., G.S.A. and J.G.S.d.S.; resources, J.G.S.d.S.; data curation, A.C.S.d.S., G.S.A. and J.G.S.d.S.; writing—original draft preparation, A.C.S.d.S.; writing—review and editing, A.C.S.d.S., G.S.A. and J.G.S.d.S.; visualization, G.S.A. and J.G.S.d.S.; supervision, G.S.A. and J.G.S.d.S.; project administration, G.S.A. and J.G.S.d.S.; funding acquisition, J.G.S.d.S. All authors have read and agreed to the published version of the manuscript.

**Funding:** This research was funded by CNPq, CAPES and FAPERJ grant number 2018.03905.3. The Article Processing Charges (APC) of 0 CHF was applied to this article, based on a MDPI discount.

**Data Availability Statement:** The data that support the findings of this study are available from the corresponding author, Santos da Silva, J.G., upon reasonable request.

**Acknowledgments:** The authors gratefully acknowledge the financial support for this research work provided by the Brazilian Science Foundation's CNPq, CAPES and FAPERJ (doctoral scholarship no. 2018.03905.3).

**Conflicts of Interest:** The authors declare no conflict of interest.

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
