# Peer review of "Advances in Methodology for Fatigue Assessment of Composite Steel–Concrete Highway Bridges Based on the Vehicle–Bridge Dynamic Interaction and Pavement Deterioration Model"

_metals, doi:10.3390/met13081343_

Round 1

Reviewer 1 Report

This study intends to assess fatigue response of concrete-steel bridges subjected to vehicle variable amplitude loads including the pavement progressive damage. The computational tool VBI was employed to evaluate overall damage resulted from the bridge-structure dynamic interaction and the pavement progressive deterioration. Damage framework has been developed over the five main steps to analyze (i) bridge-vehicle, mass and stiffness matrices through FEM, (ii) boundary conditions and critical regions/stress raisers experiencing high stress concentration factors, (iii) dynamic analysis of vehicle-bridge considering both non-deterministic irregularities and pavement deterioration over length of time, (iv) the road surface roughness and its evaluation with ISO 8608, and  (v) fatigue life assessment based on S-N curves/ Minor’s rule. The manuscript is very comprehensive and includes many variables in fatigue assessment of concrete-steel and its interaction with vehicle loading and pavement deterioration. Results are solid and valuables. Sketches are given in details and figures and charts are presented in full. The manuscript includes valuable results and it is recommended for publication in the special issue in Metals. There are a few minor comments to be addressed before the manuscript gets published in Metals.

1.     The present article is very lengthy and written with a format of a thesis. The subject matter is very comprehensive and can be shortened to a smaller article and a more focused one.

2.     RRCs are given with a general classification in Table 3. Author may elaborate that how "good" or "poor" was evaluated in design of bridges.

3.     Authors have employed Minor's rule as fatigue damage accumulation method. equation (18) has discussed after flowchart in figure 3. This needs to be restructured.

4.     Authors need to partition overall fatigue damage on composite structure. This helps the reader to identify damage on concrete, steel bars, and pavement. Authors may add a typical figure to present the weight of damage in each of these materials and their interactions for a given loading spectrum.

5.     Minor's rule linearly accumulates damage cycle-by-cycle. This approach does not include the effect of loading sequence. To assess fatigue damage in bridges undergoing random loading spectra, the inclusion of load sequence effect becomes crucial. Authors need to justify this in the manuscript.

6.     Figure 15 is not conventionally presented text in “tables/columns”. I leave this with the journal as to check the format of presentation.

7.     Conclusion section is long and need to be focused and condensed to important remarks.

The manuscript is well-written in English.

Author Response

Reviewer #1

1) The present article is very lengthy and written with a format of a thesis. The subject matter is very comprehensive and can be shortened to a smaller article and a more focused one.
The authors are very grateful for this comment. In this particular case, some modifications have been made to the introduction and to the conclusion (revision also proposed by Reviewer #1 in point 7), in order to synthesise it. In addition, synthetisations were made in the fatigue related questions (see item 2.1.5.: Step 5: Fatigue strength model in the revised version of the article). Consequently, the article has become smaller. Done!

2) RRCs are given with a general classification in Table 3. Author may elaborate that how "good" or "poor" was evaluated in design of bridges.
The authors would like to thank for the comment. The value of the RRC is calculated according to Eq. (15) and depends directly on the IRIt [calculated according to Eq. (16)]. After performing the calculation, Table 3 can be consulted to evaluate the road-roughness classification, according to ISO 8608. If the RRC value is between 8 × 10-6 to 32 × 10-6, then the pavement is classified as good. If the RRC value is between 128 × 10-6 to 512 × 10-6, then the pavement is classified as poor. Done!

3) Authors have employed Minor's rule as fatigue damage accumulation method. equation (18) has discussed after flowchart in figure 3. This needs to be restructured.
The authors fully agree with the reviewer observation. This restructuring has been carried out and can be confirmed on pages 8 and 9, item 2.1.5, in the revised version of the paper. Done!

4) Authors need to partition overall fatigue damage on composite structure. This helps the reader to identify damage on concrete, steel bars, and pavement. Authors may add a typical figure to present the weight of damage in each of these materials and their interactions for a given loading spectrum.
The authors are very grateful for this comment. However, the authors would like to emphasise that the presented methodology addresses fatigue damage on the steel. Therefore, for the consideration of damage on the concrete slabs and pavement, a different analysis methodology should be proposed which does not fit in this particular research work developed by the authors. Done!

5) Minor's rule linearly accumulates damage cycle-by-cycle. This approach does not include the effect of loading sequence. To assess fatigue damage in bridges undergoing random loading spectra, the inclusion of load sequence effect becomes crucial. Authors need to justify this in the manuscript.
The authors are very grateful for this comment. It is important to highlight that the influence of the load sequence effects and of the stress range cycles below the Constant Amplitude Fatigue Limit (CAFL) on the fatigue, life is still under debate, and represents a vast field, which is out-of-the-scope of the present research work. However, it was justified in the text that for load histories which arise from stationary processes, such as the traffic on bridges, the random occurrence of high and low stresses contributes to reducing the impact of the load sequence effects on the fatigue life (see page 8, item 2.1.5 and also see references [37] and [38], in the revised version of the paper). Done!

6) Figure 15 is not conventionally presented text in “tables/columns”. I leave this with the journal as to check the format of presentation.
The authors would like to thank for the comment. Therefore, several modifications have been made in this figure, which in the revised version of the article is referenced as Table 8 (see page 18). Done!

7) Conclusion section is long and need to be focused and condensed to important remarks.
The authors fully agree with the reviewer observation. Some modifications have been made to the conclusion in order to synthesise it (see page 22, item 7, in the revised version of the paper). Done!

Yours Sincerely

Prof. José Guilherme Santos da Silva, D.Sc.
Structural Engineering Department
Faculty of Engineering, FEN
State University of Rio de Janeiro, UERJ
E-mail: [email protected] or [email protected]

Reviewer 2 Report

This paper developed an analysis methodology to assess the dynamic structural behaviour and the fatigue behaviour of steel-concrete composite highway bridges including the vehicles-structure interaction and the pavement progressive deterioration effect. The different response parameters including the displacements and the hot-spot stress were obtained in two details of the stiffener welded to the web. Some revisions are needed before consideration of publication in the journal.

1. Line 304-305: The author stated the thickness of the concrete slab is 0.225m, However, the thickness is variable in the transverse direction as show in Fig. 4.

2. Please explain how to determine the dimension size of the sub-models shown in Fig. 13 to better avoid any errors due to the interactions.

3. Is there any guidelines to determine the mesh size at the web-gap position as shown in Fig. 14, and what is the influence of the mesh size on the results?

4. As the proposed methodology takes the vehicles-structure interaction and the pavement progressive deterioration effects into account, the comparison between the results predicted by the proposed method and those without consideration of the above-mentioned effects should be carried out and thus, the effects of these factor can be revealed.

5. The description shown in Fig. 15 is too long, and only some key factors from the comparison are recommended to illustrate.

The English presentation of this paper is fine, only minor editing of English language required.

Author Response

Reviewer #2

1) Line 304-305: The author stated the thickness of the concrete slab is 0.225m; However, the thickness is variable in the transverse direction as show in Fig. 4.
The authors would like to thank for the comment. In this case, extensive parametric analyses were performed by the authors in this same structural model, in previous work: “Fatigue life evaluation of a composite steel-concrete roadway bridge through the hot-spot stress method considering progressive pavement deterioration”, Engineering Structures, 2018, vol. 166, pp. 46–61 (see Reference [31] in the revised version of the article). Therefore, it was concluded that in this analysed structural model, the influence of the variable thickness is irrelevant for the dynamic structural analysis and for obtaining the bridge global stresses. Done!

2) Please explain how to determine the dimension size of the sub-models shown in Fig. 13 to better avoid any errors due to the interactions.
The authors would like to thank for the comment. In order to determine the dimension size of the sub-models, sensitivity analyses and numerical convergence tests were performed and validated in a previous paper published by the authors: “Fatigue life evaluation of a composite steel-concrete roadway bridge through the hot-spot stress method considering progressive pavement deterioration”, Engineering Structures, 2018, vol. 166, pp. 46–61 (see Reference 31 in the revised version of the article). Done!

3) Is there any guidelines to determine the mesh size at the web-gap position as shown in Fig. 14, and what is the influence of the mesh size on the results?
The authors are very grateful for this comment. In order to determine the mesh size at the web-gap position, the rules indicated in “Recommendations for Fatigue Design of Welded Joints and Components - IIW” for a fine mesh and a quadratic extrapolation with three extrapolation points were followed (see Reference [33] in the revised version of the paper). Thus, the developed finite element mesh of the sub model adopts solid elements with quadratic shape functions. Done! 

4) As the proposed methodology takes the vehicles-structure interaction and the pavement progressive deterioration effects into account, the comparison between the results predicted by the proposed method and those without consideration of the above-mentioned effects should be carried out and thus, the effects of these factor can be revealed.
The authors would like to thank for the comment. However, justified that since the middle 80’s the assessment of the dynamic effects due to vehicles traffic on bridge irregular pavement surfaces is investigated and numerous studies have presented the comparison between results considering and not considering the irregular surface pavement condition on the bridge decks. Therefore, the focus of this research work was not to present the difference between the results considering the non-deteriorated and the deteriorated situation, but to show the influence of progressive deterioration over time. Done!

5) The description shown in Fig. 15 is too long, and only some key factors from the comparison are recommended to illustrate.
The authors would like to thank for the comment. In this particular case, the authors have understood the reviewer comment, but would like to justify that the information was presented to facilitate the reader to fully understand the contribution proposed in this paper. However, it is important to note that modifications have been made in Figure 15 based on a comment made by the Reviewer #1. Thus, in the revised version of the article, this figure is referenced as Table 8 (see page 18). Done!

Yours Sincerely

Prof. José Guilherme Santos da Silva, D.Sc.
Structural Engineering Department
Faculty of Engineering, FEN
State University of Rio de Janeiro, UERJ
E-mail: [email protected] or [email protected]

Round 2

Reviewer 2 Report

The paper can be accepted in the present form. 

The language in this paper is fine.